# The Effects of Different Cotton Varieties on the Growth and Feeding Preferences of *Helicoverpa armigera*

**DOI:** 10.3390/insects16020115

**Published:** 2025-01-24

**Authors:** Lei Yue, Xuehua Shao, Jianhao Dong, Guoyi Xia, Xiang Yan, Aizimaitijiang Tuerxun, Wei Lu

**Affiliations:** 1Key Laboratory of the Pest Monitoring and Safety Control of Crops and Forests of the Universities of the Xinjiang Uygur Autonomous Region, College of Agronomy, Xinjiang Agricultural University, Urumqi 830052, China; yuelei0418@outlook.com (L.Y.); dong371014@163.com (J.D.); 15765637585@163.com (G.X.); yxxxx0000@163.com (X.Y.); 13390793631@163.com (A.T.); 2Engineering Research Centre of Cotton, Ministry of Education, Urumqi 830052, China; 3Guangdong Province Key Laboratary of Tropical and Subtropical Fruit Tree Research, Institute of Fruit Tree Research, Guangdong Academy of Agricultural Sciences, Guangzhou 510640, China; sxh19831017@163.com; 4Key Laboratory of South Subtropical Fruit Biology and Genetic Resource Utilization, Ministry of Agriculture and Rural Affars, Institute of Fruit Tree Research, Guangdong Academy of Agricultural Sciences, Guangzhou 510640, China; 5Western Agricultural Research Center, Chinese Academy of Agricultural Sciences, Changji 831100, China

**Keywords:** cotton pest, feeding area, developmental duration, fecundity, *Helicoverpa armigera*

## Abstract

Xinjiang is the leading cotton producer in China, occupying a pivotal place in the national economy. At present, the primary control strategies for *Helicoverpa armigera* involve the cultivation of transgenic insect-resistant cotton. However, a multitude of investigations have indicated a mounting resistance in *H. armigera* populations against these genetically modified crops. Consequently, there is an immediate necessity to identify non-GMO cotton varieties with insect resistance for wider cultivation. The intricate relationship between herbivorous insects and their plant hosts has remained a vibrant subject of study. Nevertheless, research focusing on *H. armigera*’s feeding preferences across diverse cotton species, as well as on the influence of these cotton varieties on the insect’s growth, development, and reproductive processes, remains limited. This study employed five distinct cotton varieties, with the aim of identifying those that exhibit high resistance to *H. armigera* and have low appeal as a food source; this will not only facilitate the selection of insect-resistant cultivars, but may also provide a foundation for effective pest management strategies.

## 1. Introduction

Xinjiang is an autonomous province in northwest China. It has the largest cotton planting area and the highest rate of cotton production in China. The cotton industry in Xinjiang plays an important role in China’s national economy [1]. According to the latest data from the National Bureau of Statistics, the cotton planting area in Xinjiang was 23,693 km^2^ in 2023, accounting for 85% of China’s cotton planting area. The cotton yield in Xinjiang was 5.112 million tons, accounting for 91% of China’s cotton yield. Xinjiang has been the top cotton-producing region in China for 30 consecutive years.

*Helicoverpa armigera* (Lepidoptera; Noctuidae) is an important pest in the global cotton industry that endangers the whole growth period of cotton [2,3,4,5]. It eats young leaves at the seedling stage of cotton, affecting photosynthesis. In the bud stage, it eats buds and hinders pollination. In the green boll stage, it eats cotton bolls, induces boll rot, and reduces cotton yield and quality. In China, approximately 50% of insecticides are used for the control of *H. armigera*, and the annual global economic loss caused by this species is more than USD $5 billion [6,7]. In the unique geographical environment of Xinjiang, the high generation of *H. armigera* exhibits a particularly prominent threat to cotton production. Up to three to four generations of this species can occur per year, and the cycle of each generation is about 30 days [8], posing a serious challenge to cotton yield and quality. Wang found that the economic loss caused by *H. armigera* in Xinjiang amounted to between USD $200 million and USD $450 million [9].

Therefore, the control of *H. armigera* is a problem that cannot be ignored in cotton production. The control methods include physical control, chemical control, biological control, agricultural control, and so forth [10,11,12,13,14]. However, many studies have shown that *H. armigera* has different resistance levels to Bt toxin. Two previous studies have shown that the resistance of *H. armigera* to CrylAc toxin protein reached between 1600- and 275-fold [15,16,17]. Ali et al. measured the survival rate of *H. armigera* larvae from 0 to more than 50% using the dose-distinguishing method [18]. According to Watrud and Donegan, transgenic cotton planting increases the number of bacteria and fungi in the soil [19,20]. Wilson found that although transgenic cotton reduced the attack of lepidopteran pests, it promoted the population growth of sweet potato whitefly, *Bemisia tabaci* (Gennadius) [21]. Moreover, transgenic varieties also lead to problems such as reduced cotton fibre quality, gene drift, and food safety of cottonseed oil [22,23,24].

In recent years, research on the interaction between the feeding preferences and oviposition preferences of phytophagous insects on different varieties of crops has become a hot topic in academia. Li found that the number of sweetpotato whitefly (*Bemisia tabaci*) eggs was lowest on the cultivar ‘Jade’, whereas the cultivars ‘Gold Mine’, ‘Golden Rod’, ‘Long Tendergreen’, and ’Royal Burgundy’ supported the fewest whitefly nymphs. The numbers of potato leafhopper (*Empoasca fabae*) and tarnished plant bug (*Lygus lineolaris*) adults were the lowest on the cultivars ‘Greencrop’ and ‘PV-8570’ [25]. The snap bean of the ‘Refugee’ type was demonstrated to be less heavily populated by *E*. *fabae* than the cultivar ‘Green Pod’ in Ohio [26].

At present, studies on the feeding preferences of *H. armigera* for different varieties of cotton, and the effects of different varieties of cotton on the growth and reproduction of *H. armigera*, are scarce. Therefore, this study aimed to investigate the feeding preferences of *H. armigera* for the non-transgenic cotton varieties in Xinjiang, and the influence of these varieties on the growth and development of this pest. The study also screened non-transgenic cotton varieties with high resistance or low feeding appeal to *H. armigera*, which is beneficial to the breeding of insect-resistant varieties, providing a scientific basis to reduce pest attacks.

## 2. Materials and Methods

### 2.1. Test Insect Source

A colony of *H. armigera* was provided by the Henan Jiyuan Baiyun Industry Co., Ltd., Jiyuan, China. It was reared in an incubator at constant temperature and humidity using an artificial diet. The main components of the *H. armigera*’s diet were as follows: soybean powder 80 g, wheat bran 150 g, yeast powder 30 g, agar 20 g, sucrose 20 g, casein 40 g, compound vitamin 8 g, ascorbic acid 3 g, glacial acetic acid 4 g, sorbic acid 3 g, and distilled water 1500 mL. The feeding method suggested by Liang was used in this study [27,28,29,30]. The conditions were as follows: temperature 26 ± 1 °C, relative humidity 70% ± 10%, and photoperiod light:dark = 14 h:10 h. The following rearing process was adopted: The pupae were disinfected with 10% formaldehyde solution and put into an insect cage. After eclosion, the adults were fed with 10% honey water, and white gauze was placed on the top of the insect cage for oviposition. The collected eggs were placed in a self-sealing bag and then transferred to an incubator for hatching. The environmental conditions within this incubator, including the temperature and air humidity, were congruent with those required for larval rearing. After hatching, the larvae were transferred into a 48-well plate containing the artificial diet using a fine brush, and covered with a moulded plate (with a well size of 1.2 cm × 1.2 cm × 2.0 cm). A group of five boxes were bundled with rubber bands to prevent escape. After the larvae had grown to the diet third instar, they were transferred to a 24-well insect box containing the artificial feed, and reared individually until pupation (with a well size of 2.0 cm × 2.0 cm × 2.5 cm). The tools and utensils used for insect rearing were strictly disinfected before and after use. The names and sources of the experimental cotton varieties are presented in Table 1.

### 2.2. Test Cotton Varieties

Five tested cotton varieties were planted in flowerpots in greenhouses, with 10 pots for each variety. Timely watering and fertilizing, the application of mepiquat chloride, and restricted use of pesticides, herbicides, and fungicides were ensured throughout the cotton growth period, to protect the tested cotton plants from mechanical damage, pests, diseases, and nutrient, water, and drought stress. The names of the cotton varieties above were replaced by the experimental labels A, B, C, D, and E, respectively.

### 2.3. Experimental Instruments

This study utilized the following experimental apparatus: an FB224 electronic balance (accurate to 1/10,000), manufactured by Shanghai Sunny Hengping Scientific Instrument Co., Ltd. (Shanghai, China); an SPX-250B-G microcomputer-controlled light incubator from Shanghai Boxun Medical Biological Instrument Corp. (Shanghai, China); a stereomicroscope (model MZ101) produced by Guangzhou Mingmei Optoelectronic Technology Co., Ltd. (Guangzhou, China); a SX-MD16E-2 backpack electric sprayer with a 16 L capacity, from Shixia Holdings Co., Ltd. (Taizhou, Zhejiang, China); and a pipette from Sartorius Biohit Liquid Handling Oy (Helsinki, Finland).

### 2.4. Experimental Methods

#### 2.4.1. Feeding Preferences of Larvae

The feeding preferences of *H. armigera* larvae for five different cotton varieties were determined by selective and non-selective tests.

(1) Selective feeding preference

The leaf disc method was adopted for this experiment [31]. Leaf discs with a diameter of 1.3 cm were cut out from only the middle part of young leaves (excluding the leaf base and tip). Before making the leaf discs, the dust on the leaf surface was cleaned with distilled water, the water droplets on the surface were absorbed with clean filter paper, and the leaves were dried at room temperature. During the test, a Petri dish with a diameter of 12.0 cm was selected, and a layer of wet filter paper was placed at the bottom. The leaf discs from CCRI 49, Chuangmian 508, Tahe 2, J206-5, and Zhongzhimian KV2 were placed alternately and equidistantly; 2 leaf discs of each cotton variety were used for this, with a total of 10 discs. Larvae that had been starved for 4 h were placed in the middle of each Petri dish, at an equal distance from the leaves. A total of 10 larvae of *H. armigera*, each weighing 6–8 mg and at the second instar stage, were used for each experiment, with five replicates conducted. The mouth of the dish was sealed with a plastic film with several holes, sealed with a sealing film, and placed in an incubator under dark conditions. During the test, it was observed over time that the leaf disc was completely eaten, and a new leaf disc was supplemented. After 12 h, the larvae of *H. armigera* were removed, and the leaf discs were spread on a glass plate. A mark was made, and a camera was used to take pictures of the leaves; then, the feeding area was calculated through Adobe Photoshop 2020 software, version 21.0.1.

(2) Non-selective feeding preferences

The preparation method for the leaf discs was the same as described in the selective feeding preferences section. A wet filter paper was laid on the bottom of a glass culture dish with a diameter of 12.0 cm, and six leaf discs of the same cotton variety were placed along the circumference, ensuring equal spacing between them. Larvae that had been starved for 4 h were placed in the middle of each Petri dish, at an equal distance from the leaves. A total of 20 larvae of *H. armigera*, each weighing 6–8 mg and at the second instar stage, were used for each experiment, with three replicates conducted [32]. The mouth of the dish was sealed with a plastic film with several holes, sealed with a sealing film, and placed in an incubator under dark conditions. During the test, it was observed that the leaf disc was completely eaten, and a new leaf disc was supplemented. After 12 h, the larvae of *H. armigera* were removed, and the leaf discs were spread on a glass plate. A mark was made, and a camera was used to take pictures of the leaves; then, the feeding area was calculated through Adobe Photoshop 2020 software.

(3) The mortality of *H. armigera* after feeding on different varieties of cotton leaves for 20 days.

The leaf feeding method was adopted for this experiment. Healthy second-instar larvae of *H. armigera*, each with a body weight of 6–8 mg, were selected and placed in a 24-well insect box. After starvation for 4 h, fresh leaves of the five different cotton varieties were collected and cut into leaf discs, using a puncher, to feed to the *H. armigera* larvae. Fresh leaves were added over time, based on the feeding requirements of larvae. Each cotton variety was subjected to five replicate experiments, with each replicate consisting of 20 larvae. After 20 days, the surviving *H. armigera* were counted and the mortality rate was calculated.

#### 2.4.2. The Effects of Different Cotton Varieties on the Growth, Development, and Reproduction of *H. armigera*

(1) Effects on the growth and development of *H. armigera*

The experimental method was the same as described in Section 2.4.1 [3]. The instar of *H. armigera* larvae in each group was observed and recorded every 24 h. The weight of *H. armigera* larvae was recorded on the day when the *H. armigera* entered a new instar [33]. The average body weight and developmental duration *H. armigera* larvae for each instar were calculated. The prepupal duration, pupal weight, developmental duration, pupation number, pupal malformation number, female pupal number, eclosion number, and adult malformation number were recorded. The, the pupation rate, pupa malformation rate, female ratio, eclosion rate, and adult malformation rate were calculated [34,35].

(2) Effects on the reproduction of *H. armigera*

After the pupae emerged as adults, a pair of adults (a female and a male) were randomly selected and placed in a transparent 100 mL plastic beaker, and they were fed separately with 10% honey water. Each treatment group consisted of 3 replicates, with each replicate comprising 5 pairs of male and female adults. Disinfected white gauze was placed over the cup openings for oviposition. The survival of male and female adults was observed every 24 h, and the gauze was replaced. The daily oviposition of each female was observed and recorded using a stereomicroscope. A total of 300 eggs were randomly selected from each treatment group (100 eggs per replicate, 3 replicates for each treatment group) and cultured in self-sealing bags under controlled conditions in a constant-temperature incubator (temperature 26 ± 1 °C, relative humidity 70 ± 10%, and photoperiod light:dark = 14 h:10 h). The hatching status was observed, and the number of hatched eggs was recorded. Finally, the longevity of male and female adults, the number of eggs laid by each female adult, and the hatching rate of the eggs were calculated [36,37].

### 2.5. Data Handling

#### 2.5.1. Analysis of Feeding Selection Response Data of Larvae

(1) Feeding area calculation and analysis method

Adobe Photoshop 2020 was used to calculate the feeding area [38]. After the preliminary analysis of the data through the calculation formulas described in the article, the SPSS 20.0 software was used to perform one-way analysis of variance, and the Duncan test was used for multiple comparisons to determine whether there were significant differences in multiple sets of samples. Then, Origin 2017 software was used to draw the analysis results. Different lowercase letters indicated that there were significant differences among different varieties according to Duncan’s multiple comparison (*p* < 0.05).(1)$$S=S_{1}-S_{2}$$

S represents the feeding zone for pests, with S_1_ indicating the total leaf surface area and S_2_ representing the remaining area after infestation.

(2) Mortality calculation method(2)$$Mortality=\frac{Number\;of\;death\;insects\;after\;20\;days\;of\;continuous\;feeding\;on\;cotton\;leaves}{Total\;number\;of\;testinsects\;per\;treatment}\times 100\%$$

#### 2.5.2. Calculation Method for Growth, Development and Reproduction Related Indicators of *H. armigera*

SPSS 20.0 software was used to analyze the developmental duration, weight, prepupal duration, pupation rate, pupal malformation rate, female ratio, male and female pupal weight, male and female pupal period, eclosion rate, adult malformation rate, oviposition amount, hatching rate, and life span of male and female adults in the five treatment groups.
(3)$$Pupation\;rate=\frac{Number\;of\;pupation\;per\;replicate}{Total\;number\;of\;test\;insects\;per\;replicate}\times 100\%$$
(4)$$Pupal\;deformity\;rate=\frac{The\;number\;of\;deformed\;pupae\;per\;repeat}{Number\;of\;pupae\;per\;replicate}\times 100\%$$
(5)$$Female\;ratio=\frac{Number\;of\;female\;pupae\;per\;replicate}{Number\;of\;pupae\;per\;replicate}\times 100\%$$
(6)$$Eclosion\;rate=\frac{Number\;of\;adult\;eclosion\;per\;replicate}{Number\;of\;pupae\;per\;replicate}\times 100\%$$
(7)$$Adult\;deformity\;rate=\frac{Number\;of\;adults\;per\;duplication\;deformity}{Number\;of\;adult\;eclosion\;per\;replicate}\times 100\%$$(8)$$Hatchability=\frac{Number\;of\;eggs\;hatched\;per\;replicate}{Number\;of\;eggs\;per\;replicate}\times 100\%$$


## 3. Results

### 3.1. The Effects of Different Cotton Varieties on the Feeding Preferences and Mortality of H. armigera Larvae

The selective feeding preferences test revealed that the order of the feeding area for the five different treatment groups was A > D > C > E > B. No significant difference was observed among the A, C, and D treatment groups (*p* > 0.05), or between the B and E treatment groups (*p* > 0.05). The feeding area of the A and D treatment groups was significantly higher than that of the B and E treatment groups (*p* < 0.05). The feeding area was the largest for the A treatment group (0.4916 cm^2^) and the smallest for the B treatment group (0.3953 cm^2^). The feeding areas of the C, D, and E treatment groups were 0.4599, 0.4902, and 0.4073 cm^2^, respectively (Figure 1-I,IV; Appendix A).

The non-selective feeding preference test revealed that the feeding area of the five treatment groups was in the order A > D > C > E > B. The feeding area of the A treatment group was significantly higher than that of the other four treatment groups (*p* < 0.05). No significant differences were observed among the B, C, and E treatment groups (*p* > 0.05), or between the C and D treatment groups (*p* > 0.05). The feeding area was the largest for the A treatment group (0.5309 cm^2^) and the smallest for the B treatment group (0.4032 cm^2^). The feeding areas of the C, D, and E treatment groups were 0.4468, 0.4709, and 0.4082 cm^2^, respectively (Figure 1-II,V; Appendix A).

After continuous feeding of different varieties of cotton leaves for 20 days, the mortality rate of the five treatment groups was in the order B > E > C > D > A. A significant difference was observed between the A and B treatment groups (*p* < 0.05). The mortality rate was the highest for the B treatment group (33%) and the lowest (21%) for the A treatment group. The mortality rates of the C, D, and E treatment groups were 28%, 25%, and 31%, respectively (Figure 1-III; Appendix A).

### 3.2. The Effects of Different Cotton Varieties on the Growth and Development of H. armigera Larvae of Different Instars

The effects of different varieties of cotton leaves on the weight of each instar of *H. armigera* showed no significant differences for the third-instar larvae among the A, C, and D treatment groups (*p* > 0.05), or between the B and E treatment groups (*p* > 0.05). The weights of the larvae in the A, C, and D treatment groups were significantly higher than in the B and E treatment groups (*p* < 0.05). For the fourth instar, the significant differences in body weight between different treatment groups were the same as for the third instar, but the body weights of the fourth instar larvae in each treatment group increased. The weights of the third-instar larvae in the five treatment groups were 18.79, 15.60, 18.42, 18.59, and 16.21 mg, respectively, and those of the fourth-instar larvae were 37.93, 31.14, 35.56, 36.89, and 32.22 mg, respectively (Figure 2-I,II; Appendix A). The weights of the fifth-instar larvae were not significantly different among the A, C, D, and E treatment groups (*p* > 0.05); however, the weight of the fifth instar larvae in the B treatment group was significantly lower than that in the A, C, and D treatment groups (*p* < 0.05). The body weights of the fifth-instar larvae in the five treatment groups were 93.67, 83.93, 91.39, 92.76, and 87.60 mg, respectively (Figure 2-III; Appendix A). The weight of the sixth-instar larvae was significantly lower in the B treatment group compared with the other treatment groups (*p* < 0.05). The lowest body weight of the B treatment group was 206.03 mg, which was 58.75, 33.70, 45.00, and 16.16 mg lower than that of the other four treatment groups, respectively (Figure 2-IV; Appendix A). The insect weights, ranging from high to low in the five different treatment groups, were as follows: A > D > C > E > B.

The effects of different cotton varieties on the durations of the third to sixth instar larvae of *H. armigera* revealed no significant differences among the A, C, and D treatment groups (*p* > 0.05), or between the B and E treatment groups (*p* > 0.05). The duration in the B treatment group was the longest, and was significantly higher than that in the A and D treatment groups (*p* < 0.05). Compared with the A treatment group, the duration of the third- to sixth-instar larvae in the B treatment group increased by 0.64, 1.20, 0.84, and 0.70 days, respectively. Compared with the D treatment group, the durations of the third- to sixth-instar larvae in the B treatment group were increased by 0.63, 0.87, 0.81, and 0.62 days, respectively (Figure 2-V,VIII; Appendix A). The duration of the third-instar larvae in the B treatment group was 6.43 days, which was 0.64, 0.54, 0.63, and 0.16 days higher than that of the other four treatment groups, respectively. The duration of the fourth-instar larvae in the B treatment group increased by 1.20, 0.72, 0.87, and 0.33 days compared with the other four treatment groups, respectively. The duration of the fifth-instar larvae in the B treatment group increased by 0.84, 0.55, 0.81, and 0.17 days compared with the other four treatment groups, respectively. The duration of the sixth-instar larvae in the B treatment group increased by 0.70, 0.49, 0.62, and 0.20 days compared with the other four treatment groups, respectively.

### 3.3. The Effects of Different Cotton Varieties on the Pupae of H. armigera

The effects of the five different cotton varieties on the prepupal duration of *H. armigera* revealed no significant differences among the A, C, and D treatment groups (*p* > 0.05) or between the B and E treatment groups (*p* > 0.05). The effects on the A, C, and D treatment groups were significantly lower than those on the B and E treatment groups (*p* < 0.05). The prepupal period in the B treatment group was the longest (6.19 days), which was 0.75, 0.58, 0.65, and 0.15 days higher than that in the other four treatment groups (Figure 3-I; Appendix A). The pupation rates in the five treatment groups were 79%, 67%, 72%, 75%, and 69%, respectively. The pupation rate in the A treatment group was significantly higher than that in the B and E treatment groups (*p* < 0.05, Figure 3-II; Appendix A). The pupal malformation rates were 20.21%, 30.13%, 22.37%, 21.23%, and 29.13% in the five treatment groups, respectively. The pupal malformation rates in the B and E treatment groups were significantly higher than those in the A, C, and D treatment groups (*p* < 0.05, Figure 3-III; Appendix A). Malformation of the pupa is mainly manifested as abnormal moulting or incomplete moulting, and results in a shrivelled pupa. The proportion of female insects within each treatment cohort was 51.84%, 49.24%, 51.35%, 52.18%, and 49.04%, respectively, with no statistically significant variations (*p* > 0.05, Figure 3-IV; Appendix A).

The results of the pupal weight in the five treatment groups revealed no significant differences in the weights of the female pupae and male pupae between the A and D treatment groups (*p* > 0.05), or between the B and E treatment groups (*p* > 0.05). The pupal weight in the B treatment group was the lowest, with 203.76 mg for female pupa and 193.25 mg for male pupa. The pupal weight in the A treatment group was the highest, with 252.84 mg for female pupa and 245.76 mg for male pupa (Figure 3-V,VI; Appendix A). The results of pupal duration across the five treatment groups revealed no significant differences in female pupal duration among the B, C, and E treatment groups (*p* > 0.05). However, the pupal duration in these groups was significantly higher than in the A and D treatment groups (*p* < 0.05). The A treatment group had the shortest duration of 8.74 days, and the B treatment group had the longest duration of 10.07 days (Figure 3-VII; Appendix A). The male pupal duration in the B treatment group was significantly higher than that in the other four treatment groups (*p* < 0.05). Among these, the male pupal duration was the shortest in the A treatment group (8.94 days) and the longest in the B treatment group (10.49 days) (Figure 3-VIII; Appendix A).

### 3.4. The Effects of Different Cotton Varieties on the Reproduction of H. armigera

The eclosion and adult malformation rates of *H. armigera* were significantly different between the B and E treatment groups and among the A, D, and E treatment groups (*p* < 0.05). Among these, the A treatment group had the highest eclosion rate (73.46%) and the lowest adult malformation rate (29.28%). The B treatment group had the lowest eclosion rate (55.27%) and the highest adult malformation rate (42.70%) (Figure 4-I,II; Appendix A). The results of oviposition and egg hatching showed no significant differences in oviposition between the five treatment groups (*p* > 0.05), with the average number of ovipositions by each female in each treatment group equaling 628.67, 532.80, 584.60, 593.27, and 544.13 eggs, respectively. No significant differences in hatching rate were found between the B and E treatment groups (*p* > 0.05); however, the hatching rates in these groups were significantly lower than those in the other treatment groups (A > D > C > E > B) (Figure 4-III,IV; Appendix A). The results of female and male adult longevity revealed no significant difference in female longevity between the A and D treatment groups (*p* > 0.05), and it was significantly higher than that in the B and E treatment groups (*p* < 0.05). Among these, the B treatment group had the shortest lifespan (8.12 days), which was shortened by 1.08, 0.71, 0.9, and 0.26 days compared to the other treatment groups, respectively. There were no significant differences in male longevity observed among the A, C, and D groups (*p* > 0.05), nor between the B and E groups (*p* > 0.05). Moreover, the longevity for treatment groups A, C, and D was significantly higher than that for the treatment groups B and E (*p* < 0.05). The longevity of treatment B was the shortest (7.35 days), which was 1.59, 1.12, 1.35, and 0.19 days shorter than that of the other treatment groups, respectively (Figure 4-V,VI; Appendix A).

## 4. Discussion

The interaction between herbivorous insects and plants has always been a hot topic for researchers [39,40,41,42,43,44,45]. However, in previous studies, the effects of the CCRI 49, Chuangmian 508, Tahe 2, J206-5, and Zhongzhimian KV2 cotton varieties on the feeding preference and growth and development of *H. armigera* have not been reported. In the process of studying the effects of CCRI 49 on the growth, development, and reproduction of *H. armigera*, researchers such as Zhang have found a remarkable phenomenon: the survival rate of newly hatched *H. armigera* larvae after feeding on CCRI 49 leaves is less than 30% [46]. This important discovery not only shows that the plant has a significant inhibitory effect on the larvae of *H. armigera*, but also provides key data on the larval development cycle of 22.99 days, the pupal stage of 12.87 days, the pupal weight of up to 208.60 mg, and the emergence rate of 84.43%. These data provide strong evidence for an in-depth understanding of the ecological links between *H. armigera* and plants. On this basis, this study conducted an in-depth discussion, and obtained some results that are different from previous studies, but also have implications. We observed that the mortality rate of the late second-instar larvae of *H. armigera* reached 21% after feeding on CCRI 49 leaves for 20 days. Although slightly lower than the results of researchers such as Zhang, this still indicates that this plant has a significant lethal effect on *H.armigera* larvae. In addition, the developmental cycles of larvae of different growth stages were recorded in detail. The developmental cycles of 3–6-instar larvae were 5.79 days, 5.16 days, 5.28 days, and 4.37 days, respectively. In contrast to Zhang’s study, we only recorded the duration of 3–6 instars, but through our detailed records of the duration of each instar, we revealed changes in the development rate of larvae in continuous instars. In terms of gender differences, this study found that the male and female pupal periods were 8.74 days and 8.94 days, respectively, and the male and female pupal weights were 252.84 mg and 245.76 mg, respectively. These results suggest that although there are differences in development time and body weight between male and female pupae, the differences are not significant, and may not have a significant impact on the subsequent adult stage. In addition, compared with Zhang’s study, in our study, the pupal period was shortened, but the pupal weight was increased. However, it is worth noting that the emergence rate in our study was only 73.46%, which is lower than that reported by Zhang et al. This may be related to the different larval instars selected in the experiment. First, the selected larvae may have different levels of physiological maturity and energy reserves at early or late stages of development, affecting their ability to successfully pupate and become adults. Secondly, compared with Zhang’s research, the environmental conditions provided at each larval stage (such as temperature, humidity, and food supply) were different, which may have affected their emergence rate. In addition, genetic differences or adaptation within larval populations used in different studies may also play a role in determining eclosion rates. In addition, Wu found that the feeding preference rate of *H. armigera* for the leaves of CCRI 49 was 80.33% [47], which is consistent with the feeding preference observed in this study by selective and non-selective feeding (the feeding areas were 0.4916 and 0.5309 cm^2^, respectively). This further confirms the special position of CCRI 49 in the feeding preferences of *H. armigera*. At the same time, Zhang found that the weights of *H. armigera* and beet armyworms (*Spodoptera exigua*) larvae fed on the leaves of three cotton varieties [48], Mianmei 9101, CCRI 12, and Shiyuan 321, were lower than the weight of those fed on the leaves of Lumian 12, CCRI 22, and Giza 77, suggesting that different insects may have similar feeding preferences.

Moreover, other scholars have also studied the effects of other crops on the feeding preferences and growth and development of phytophagous pests. Liu found significant differences in the feeding preferences of *Spodoptera frugiperda* larvae for different wheat varieties [49]. The larvae preferred varieties such as HM33 and fed less on LY502, and feeding different wheat varieties affected the growth and development of *S. frugiperda* larvae. Yang found that different rice varieties affected the survival rate and adult longevity of *Sesamia inferens* larvae [50]. He also found that different varieties of pepper leaves affected the growth and mortality of *S. litura* larvae [51]. The aforementioned studies have emphasized the key role of crop varieties in the prevention and control of phytophagous insects. In summary, the feeding preferences of phytophagous insects for different varieties of crops is a common phenomenon, and this preference often has a profound impact on the growth, development, and reproduction of insects.

This study had some shortcomings. First, only five varieties were selected as the research objects in this study. However, many cotton varieties are planted in China, and other varieties should be added in future experiments. Second, why *H. armigera* had a feeding preference for different varieties of cotton, and the differences in their growth, development, and reproduction, should be investigated to further explore the performance of Chuangmian 508 in different ecological environments and the molecular mechanisms of insect resistance. Finally, only greenhouse and indoor experiments were carried out in this study, but field experiments have more complex influencing factors. Therefore, field experiments should be conducted in different regions of Xinjiang in the future to provide better scientific theoretical guidance for the prevention and control of *H. armigera* and resistance control in Xinjiang.

## 5. Conclusions

In this study, we found significant differences in the feeding preferences of five different cotton varieties towards *H*. *armigera*, as well as in their impacts on the insect’s growth, development, and reproduction. Specifically, *H. armigera* had the highest feeding preference for CCRI 49, while Chuangmian 508 showed strong insect resistance. The mortality rate of *H. armigera* fed on Chuangmian 508 was higher; the larval weight, pupation rate, eclosion rate, egg hatching rate, and adult longevity were lower; the duration of each instar was prolonged; and the deformity rate of pupae and adults was higher. In addition, we have added new findings that the type of cotton variety has no significant effect on the sex ratio of *H. armigera*.

## Figures and Tables

**Figure 1 insects-16-00115-f001:**
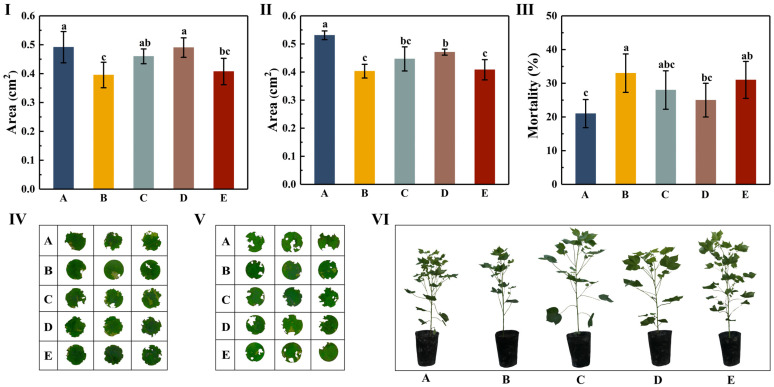
Feeding area and mortality of *H. armigera* larvae by different cotton varieties. (**I**) In selective experiment, feeding area of *H. armigera* for 5 varieties of cotton (mean ± standard deviation). (**II**) In non-selective experiment, feeding area of *H. armigera* for 5 varieties of cotton (mean ± standard deviation). (**III**) Mortality of *H. armigera* after feeding on different varieties of cotton leaves for 20 days (mean ± standard deviation). (**IV**) Leaf disc photos of selective feeding test of *H. armigera* on 5 varieties of cotton. (**V**) Leaf disc photos of non-selective feeding test of *H. armigera* on five varieties of cotton. (**VI**) Five different varieties of cotton plant photos. Different lowercase letters indicate significant differences among different varieties according to Duncan’s multiple comparison (*p* < 0.05). Uppercase letters A, B, C, D and E in figure represent CCRI 49, Chuangmian 508, Tahe 2, J206-5, and Zhongzhimian KV2, respectively.

**Figure 2 insects-16-00115-f002:**
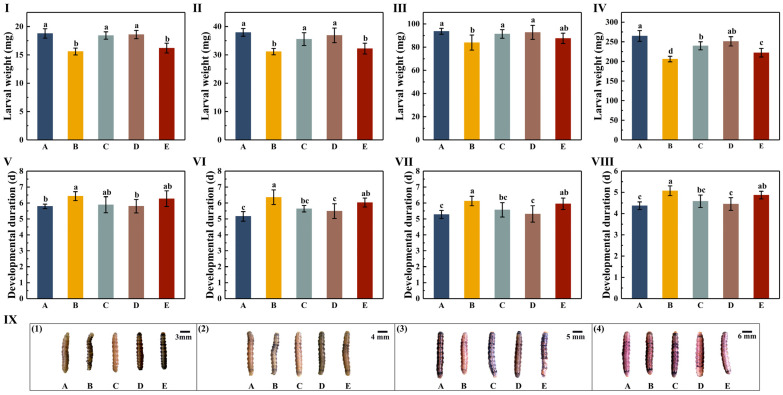
The effects of different cotton varieties on the body weight and duration of each instar of *H. armigera* larvae. (**I**) The weights of the 3rd-instar larvae of *H. armigera* after feeding on five varieties of cotton (mean ± standard deviation). (**II**) The weights of the 4th-instar larvae of *H. armigera* after feeding on five varieties of cotton (mean ± standard deviation). (**III**) The weights of the 4th-instar larvae of *H. armigera* after feeding on five varieties of cotton (mean ± standard deviation). (**IV**) The weights of the 6th-instar larvae of *H. armigera* after feeding on five varieties of cotton (mean ± standard deviation). (**V**) The durations of 3rd-instar *H. armigera* after feeding on 5 varieties of cotton (mean ± standard deviation). (**VI**) The durations of 4th-instar *H. armigera* after feeding on 5 varieties of cotton (mean ± standard deviation). (**VII**) The durations of 5th-instar *H. armigera* after feeding on 5 varieties of cotton (mean ± standard deviation). (**VIII**) The durations of 6th-instar *H. armigera* after feeding on 5 varieties of cotton (mean ± standard deviation). (**IX-1**) Photos of the 3rd-instar larvae of *H. armigera* in the 5 treatment groups. (**IX-2**) Photos of the 4th-instar larvae of *H. armigera* in the 5 treatment groups. (**IX-3**) Photos of the 5th-instar larvae of *H. armigera* in the 5 treatment groups. (**IX-4**) Photos of the 6th-instar larvae of *H. armigera* in the 5 treatment groups. Different lowercase letters indicate that there were significant differences among different varieties according to Duncan’s multiple comparison (*p* < 0.05). The uppercase letters A, B, C, D and E in the figure represent CCRI 49, Chuangmian 508, Tahe 2, J206-5 and Zhongzhimian KV2, respectively.

**Figure 3 insects-16-00115-f003:**
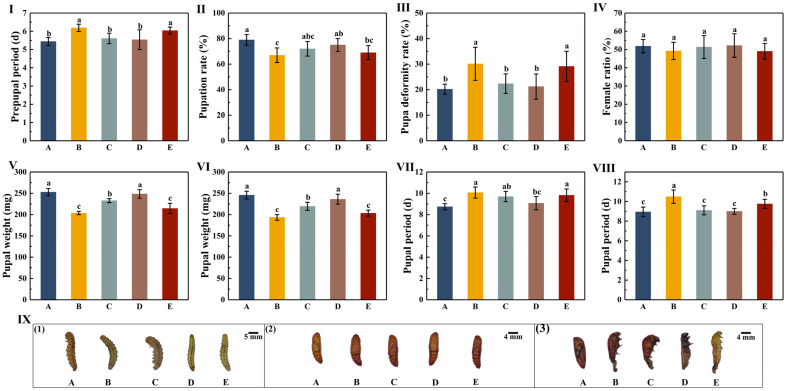
The effects of different cotton varieties on *H. armigera* pupae. (**I**) The prepupal periods of *H. armigera* after feeding on five different cottons (mean ± standard deviation). (**II**) The pupation rates of *H. armigera* after feeding on five different cottons (mean ± standard deviation). (**III**) The pupal malformation rates of *H. armigera* after feeding on five different cottons (mean ± standard deviation). (**IV**) The female ratios of *H. armigera* after feeding on five different cottons (mean ± standard deviation). (**V**) The female pupal weights of *H. armigera* after feeding on five different cottons (mean ± standard deviation). (**VI**) The male pupal weights of *H. armigera* after feeding on five different cottons (mean ± standard deviation). (**VII**) The female pupal stages of *H. armigera* after feeding on five different cottons (mean ± standard deviation). (**VIII**) The male pupal stages of *H. armigera* after feeding on five different cottons (mean ± standard deviation). (**IX-1**) *H. armigera* prepupae after feeding on five varieties of cotton. (**IX-2**) Healthy pupae of *H. armigera*. (**IX-3**) Abnormal pupae of *H. armigera*. Different lowercase letters indicate that there were significant differences among the different varieties according to Duncan’s multiple comparison (*p* < 0.05). The uppercase letters A, B, C, D, and E in the figure represent CCRI 49, Chuangmian 508, Tahe 2, J206-5, and Zhongzhimian KV2, respectively.

**Figure 4 insects-16-00115-f004:**
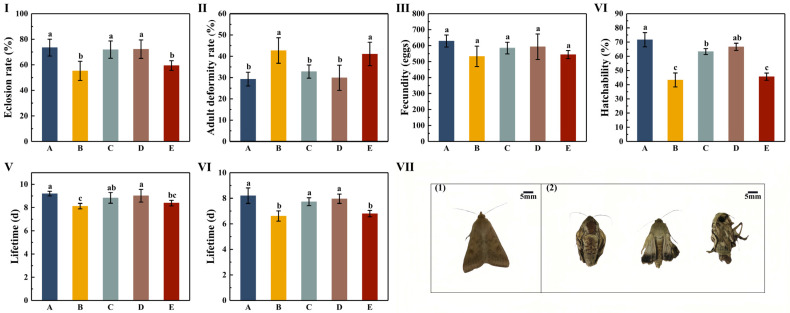
The effects of different cotton varieties on the reproduction of *H. armigera*. (**I**) Eclosion rate of *H. armigera* after feeding on five different cottons (mean ± standard deviation). (**II**) The adult deformity rate of *H. armigera* after feeding on five different cottons (mean ± standard deviation). (**III**) The number of eggs laid by *H. armigera* after feeding on five different cottons (mean ± standard deviation). (**IV**) The hatching rate of *H. armigera* after feeding on five different cottons (mean ± standard deviation). (**V**) The longevity of female adults of *H. armigera* after feeding on five different cottons (mean ± standard deviation). (**VI**) The longevity of male adults of *H. armigera* after feeding on five different cottons (mean ± standard deviation). (**VII-1**) Healthy adults of *H. armigera*. (**VII-2**) Deformed adults of *H. armigera*. Different lowercase letters indicate that there were significant differences among different varieties according to Duncan’s multiple comparison (*p* < 0.05). The uppercase letters A, B, C, D, and E in the figure represent CCRI 49, Chuangmian 508, Tahe 2, J206-5, and Zhongzhimian KV2, respectively.

**Table 1 insects-16-00115-t001:** Test cotton varieties.

Name of Test Varieties	Experiment Label	Source
CCRI 49	A	Institute of Western Agricultural, CAAS
Chuangmian 508	B	
Tahe 2	C	
J206-5	D	
Zhongzhimian KV2	E	

## Data Availability

The data presented in this study are available on request from the corresponding author.

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
