# Peer review of "The Effects of Different Cotton Varieties on the Growth and Feeding Preferences of Helicoverpa armigera"

_insects, 2025, doi:10.3390/insects16020115_

Round 1

Reviewer 1 Report

Comments and Suggestions for Authors

This manuscript represents the results of the experimental laboratory study on the relative preference and suitability of 5 cotton varieties for feeding, development, and reproduction of a dangerous pest, the cotton boll worm, Helicoverpa armigera. The experiments were well designed and conducted. Statistical analysis is correct. The results of the study can be used for the selection of the best cotton variety for commercial production. Thus, the manuscript can be published, although it needs a number of minor corrections and improvements (see below).

Lines 1-70: I do not think that such a long and detailed part of the Introduction concerning cotton is really needed in this paper. A couple of sentences would be enough to indicate the importance of cotton for the local and national economy and to list main local varieties.

Instead, I would consider interrelations of food plant preference and performance in phytophagous insects from different taxa: is it always a simple strong positive correlation? In addition, examples of similar studies conducted with other cultivated plants and other insect pests could be given here.

Lines 77-78: For the convenience of all international readers, it would be kind to convert yuans to USD (as in line 76).

Line 107: Please, indicate the conditions (temperature, air humidity, etc.) inside this incubator for egg hatching. If they were the same as for larval feeding (lines 102-103) this should be clearly stated.

Lines 107-108: Please, indicate the well size for this 48-well plate..

Line 110: Was it the same type of 24-well insect box that was used for the mortality tests (lines 170-171)? If yes, the well size should be indicated here (not below, in line 171). If not, please, indicate the well size for these boxes.

Lines 146, 159 and 169: for feeding preference tests, “same-sized second instar larvae” were used, whereas to estimate the mortality, “late second instar larvae with the same body weight” were used. Please, explain the reasons of this difference (size vs. weight) and also indicate approximated mean size and weight (or their variation ranges).

Line 152: Possibly, the leaf discs were not photographed but scanned (as in line165)? If not, why in the same line “the scanning resolution” is mentioned? If they were really photographed, why two different methods for square estimation were used for very similar tests?

Line 171: Delete superscript “3” after “cm” – size is measured in linear, not in cubic centimeters.

Lines 174-175: “Each experiment was conducted with 5 replicates per treatment, resulting in 20 replicates for each treatment” – this sentence is confusing. Possibly, some words were lost, such as “was conducted 4 times with 5 replicates”?

Lines 180-182: Was the weight of larvae recorded every 24 h (line 181) or only on the day when the larvae entered a new instar (line 182)?

Line 182-183: Incomplete sentence (the verb is absent). Possibly, it is the beginning of the next one?

Line 199: WHAT «controlled conditions”? Please, indicate clearly: temperature, humidity, etc.

Lines 218-219, equation (6): are you sure that namely “pupation rate” in percents (not the total number of pupae) was used to calculate the eclosion rate?

Lines 218-219, equations (3) – (6): What you mean “total number”: total per replicate or total per the whole experiment? This should be clearly indicated.

General comment to figures: the graphs are too small and of rather low quality. In particular, it is practically impossible to read the words and numbers along the vertical axis and the letters above the bars.

Figure 1-III – What measure of variation is shown for the percentage of mortality?

Line 257: This sentence is a bit confusing: “no significant difference between the third and fourth instar larvae” means that these two instars were of the same mean weight? This is just impossible (see lines 261-262). Possibly, it is a misprint and the comma after “larvae” should be replaced by “from” or “among”?

Figure 2: What measure of variation is shown for weight and duration? Is it SE as in the first figure”. In any case, this information should be included in the legend.

Figures 3 and 4: Again, what measure of variation is shown? Is it the same for duration (days), weight (mg), rates (%), etc.?

Line 418: Spodoptera exigua should be in the Italics font.

Line 421: I suggest that the text starting from “In summary..” should be separated in a new paragraph.

Lines 424, 425, etc.: S. frugiperda, S. inferens, etc. should be in the Italics font. Please, check carefully all Latin names in the manuscript.

Lines 433-462: This is too long text for the Conclusions that are expected to be several short but informative sentences. Please, consider to make it shorter.

Line 473: Separate the words “The corresponding”

Author Response

We sincerely thank you for your valuable comments and suggestions. In response to the problems you pointed out, we have made corresponding revisions. If there is any deviation in the process of understanding your amendment, or if the amendment content does not accurately reflect your intention, please feel free to contact us.

  1. Lines 1-70: I do not think that such a long and detailed part of the Introduction concerning cotton is really needed in this paper. A couple of sentences would be enough to indicate the importance of cotton for the local and national economy and to list main local varieties.

Instead, I would consider interrelations of food plant preference and performance in phytophagous insects from different taxa: is it always a simple strong positive correlation? In addition, examples of similar studies conducted with other cultivated plants and other insect pests could be given here.

Answer: Thank you for your valuable suggestion. We have revised the chapter on cotton in the Introduction to remove some non-core information. (Line 48-55, page 2)

Based on your suggestions, we have added examples from other studies on crops and plant-eating insects, and optimized the logical structure of the Introduction.. (Revised manuscript, Line 80-96, page 2)

  1. Lines 77-78: For the convenience of all international readers, it would be kind to convert yuans to USD (as in line 76).

Answer: Thank you for your valuable suggestion. We have converted the currency units into USD. (Revised manuscript, Line 67, page 2)

  1. Line 107: Please, indicate the conditions (temperature, air humidity, etc.) inside this incubator for egg hatching. If they were the same as for larval feeding (lines 102-103) this should be clearly stated.

Answer: Thank you for your valuable suggestion. In this study, the temperature, humidity and light conditions of all incubators were consistent, and we have clearly stated the culture conditions. (Revised manuscript, Line 110-112, page 3)

  1. Lines 107-108: Please, indicate the well size for this 48-well plate.

Answer: Thank you for your valuable suggestion. We have described the dimensions of the well in the 48-well plate. (Revised manuscript, Line 113-114, page 3)

  1. Line 110: Was it the same type of 24-well insect box that was used for the mortality tests (lines 170-171)? If yes, the well size should be indicated here (not below, in line 171). If not, please, indicate the well size for these boxes.

Answer: Thank you for your valuable suggestion. The 24-well plate sizes used in this study are all the same, and we have added a note on the 24-well plate sizes here. (Revised manuscript, Line 116-117, page 3)

  1. Lines 146, 159 and 169: for feeding preference tests, “same-sized second instar larvae” were used, whereas to estimate the mortality, “late second instar larvae with the same body weight” were used. Please, explain the reasons of this difference (size vs. weight) and also indicate approximated mean size and weight (or their variation ranges).

Answer: Thank you for your valuable suggestion. We are very sorry thatthis is a mistake in our writing. In fact, in the feeding preference and mortality, as well as in the subsequent growth, development and reproduction experiments, the second instar larvae with a body weight of 6-8mg were selected. (Revised manuscript, Line 149-152, Line 162-165, Line 176, page 4)

  1. Line 152: Possibly, the leaf discs were not photographed but scanned (as in line165)? If not, why in the same line “the scanning resolution” is mentioned? If they were really photographed, why two different methods for square estimation were used for very similar tests?

Answer: Thank you for your valuable suggestion. We are very sorry for the confusion caused to you by the wrong description of the experimental method. In fact, we first use the camera to take pictures of the leaves, and then calculate the feeding area by Adobe Photoshop 2020 software. (Revised manuscript, Line 156-158, Line 171-172, page 4)

  1. Line 171: Delete superscript “3” after “cm” – size is measured in linear, not in cubic centimeters.

Answer: Thank you for your valuable suggestion. We have implemented corresponding changes to this part of the content in the revised manuscript. (Revised manuscript, Line 114 and 117, page 3)

  1. Lines 174-175: “Each experiment was conducted with 5 replicates per treatment, resulting in 20 replicates for each treatment” – this sentence is confusing. Possibly, some words were lost, such as “was conducted 4 times with 5 replicates”?

Answer: Thank you for your valuable suggestion. We apologize for the error in the description of the relevant content. The experimental treatment should be "Each cotton variety was subjected to five replicate experiments, with each replicate consisting of 20 larvae". (Revised manuscript, Line 180-181, page 4)

  1. Lines 180-182: Was the weight of larvae recorded every 24 h (line 181) or only on the day when the larvae entered a new instar (line 182)?

Answer: Thank you for your valuable suggestion. We apologize for the error in the description of the relevant content. We observed and recorded the instars of H.armigera larvae in each group every 24 h. On the day when H.armigera entered the new instar, the weight of H.armigera larvae was recorded. (Revised manuscript, Line 186-188, page 4)

  1. Line 182-183: Incomplete sentence (the verb is absent). Possibly, it is the beginning of the next one?

Answer: Thank you for your valuable suggestion. We apologize for not being clear about the relevant content. We have made changes to the relevant content. (Revised manuscript, Line 188-193, page 4)

  1. Line 199: WHAT «controlled conditions”? Please, indicate clearly: temperature, humidity, etc.

Answer: Thank you for your valuable suggestion. We have added the control conditions to the revised manuscript. (Revised manuscript, Line 204-205, page 5)

  1. Lines 218-219, equation (6): are you sure that namely “pupation rate” in percents (not the total number of pupae) was used to calculate the eclosion rate?

Answer: Thank you for your valuable suggestion. Sorry, this is a translation error and should be calculated using the total number of pupae in each repeat. We have made changes to the relevant content. (Revised manuscript, Lines 228, equation (6), page 5)

  1. Lines 218-219, equations (3)-(6): What you mean “total number”: total per replicate or total per the whole experiment? This should be clearly indicated.

Answer: Thank you for your valuable suggestion. The formula refers to the total number of individuals in each repetition. We have modified these equations more explicitly. (Revised manuscript, Lines 228, equation (2)-(9), page 5-6)

  1. General comment to figures: the graphs are too small and of rather low quality. In particular, it is practically impossible to read the words and numbers along the vertical axis and the letters above the bars.

Answer: Thank you for your valuable suggestion. We are very sorry, we submitted manuscripts are high-definition images, but due to some unknown reasons, resulting in reduced image quality. The image has been re-uploaded to a higher definition version in the form of an attachment, and the data in the chart (mean, standard deviation, and significance) has been supplemented in the form of an attached table. (Supplementary File, page 1-5; Figures, Figure 1-4)

  1. Figure 1-III–What measure of variation is shown for the percentage of mortality?

Answer: Thank you for your valuable suggestion. We use a brush to touch the cotton bollworm gently. If it does not move, it is judged that it has died. The ordinate is the mortality rate ( % ), and the abscissa is the treatment group of different varieties of cotton, where A, B, C, D, E represent CCRI 49, Chuangmian 508,Tahe 2, J206-5, Zhongzhimian KV2, respectively. Mean, standard deviation and significance are shown in the figure. Different lowercase letters indicated that there were significant differences among different varieties by Duncan multiple comparison (P < 0.05). In the figure note section, we have made a more clear exposition. Due to previous writing errors, all charts in this article use standard deviation. If we do not understand your suggestion, please feel free to contact us. (Revised manuscript, Lines 255-264, page 6-7, Supplementary File, page 1; Figures, Figure 1)

  1. Line 257: This sentence is a bit confusing: “no significant difference between the third and fourth instar larvae” means that these two instars were of the same mean weight? This is just impossible (see lines 261-262). Possibly, it is a misprint and the comma after “larvae” should be replaced by “from” or “among”?

Answer: Thank you for your valuable suggestion. This part is due to the problem of we translation, so you are confused about this content, for which We are sorry. We have made changes to the relevant content. (Revised manuscript, Lines 267-273, page 7)

  1. Figure 2: What measure of variation is shown for weight and duration? Is it SE as in the first figure”. In any case, this information should be included in the legend.

Answer: Thank you for your valuable suggestion. We are very sorry, we submitted manuscripts are high-definition images, but due to some unknown reasons, resulting in reduced image quality. The image has been re-uploaded to a higher definition version in the form of an attachment, and the data in the chart (mean, standard deviation, and significance) has been supplemented in the form of an attached table. Mean, standard deviation and significance are shown in the figure. Different lowercase letters indicated that there were significant differences among different varieties by Duncan multiple comparison (P < 0.05). Due to previous writing errors, all charts in this article use standard deviation. If we do not understand your suggestion, please feel free to contact us. (Revised manuscript, Lines 305-321, page 8, Supplementary File, page 1-3; Figures, Figure 2)

  1. Figures 3 and 4: Again, what measure of variation is shown? Is it the same for duration (days), weight (mg), rates (%), etc.?

Answer: Thank you for your valuable suggestion. We are very sorry, we submitted manuscripts are high-definition images, but due to some unknown reasons, resulting in reduced image quality. The image has been re-uploaded to a higher definition version in the form of an attachment, and the data in the chart (mean, standard deviation, and significance) has been supplemented in the form of an attached table. Mean, standard deviation and significance are shown in the figure. Different lowercase letters indicated that there were significant differences among different varieties by Duncan multiple comparison (P < 0.05). Due to previous writing errors, all charts in this article use standard deviation. If we do not understand your suggestion, please feel free to contact us. (Revised manuscript, Lines 355-369, page 9, Supplementary File, page 3-4; Figures, Figure 3)

  1. Line 418: Spodoptera exigua should be in the Italics font.

Answer: Thank you for your valuable suggestion. We have made changes to the relevant content. (Revised manuscript, Lines 455, page 11)

  1. Line 421: I suggest that the text starting from “In summary..” should be separated in a new paragraph.

Answer: Thank you for your valuable suggestion. We have modified the logical structure of this part of the content to make it clearer. (Revised manuscript, Lines 453-464, page 11)

In addition, at the end of the discussion section of this paper, we elaborate on the limitations of this study and put forward the prospect of future research directions. (Revised manuscript, Lines 465-475, page 11-12)

  1. Lines 424, 425, etc.: S. frugiperda, S. inferens, etc. should be in the Italics font. Please, check carefully all Latin names in the manuscript.

Answer: Thank you for your valuable suggestion. We have made changes to the relevant content. (Revised manuscript, Lines 457-459, page 11)

  1. Lines 433-462: This is too long text for the Conclusions that are expected to be several short but informative sentences. Please, consider to make it shorter.

Answer: Thank you for your valuable suggestion. We make a more brief summary of the content of the conclusion. (Revised manuscript, Lines 477-485, page 12)

  1. Line 473: Separate the words “The corresponding”

Answer: Thank you for your valuable suggestion. We have made changes to the relevant content. (Revised manuscript, Lines 496-497, page 12)

  1. Lines 1-70: I do not think that such a long and detailed part of the Introduction concerning cotton is really needed in this paper. A couple of sentences would be enough to indicate the importance of cotton for the local and national economy and to list main local varieties.

Instead, I would consider interrelations of food plant preference and performance in phytophagous insects from different taxa: is it always a simple strong positive correlation? In addition, examples of similar studies conducted with other cultivated plants and other insect pests could be given here.

Answer: Thank you for your valuable suggestion. We have revised the chapter on cotton in the Introduction to remove some non-core information. (Line 48-55, page 2)

Based on your suggestions, we have added examples from other studies on crops and plant-eating insects, and optimized the logical structure of the Introduction.. (Revised manuscript, Line 80-96, page 2)

  1. Lines 77-78: For the convenience of all international readers, it would be kind to convert yuans to USD (as in line 76).

Answer: Thank you for your valuable suggestion. We have converted the currency units into USD. (Revised manuscript, Line 67, page 2)

  1. Line 107: Please, indicate the conditions (temperature, air humidity, etc.) inside this incubator for egg hatching. If they were the same as for larval feeding (lines 102-103) this should be clearly stated.

Answer: Thank you for your valuable suggestion. In this study, the temperature, humidity and light conditions of all incubators were consistent, and we have clearly stated the culture conditions. (Revised manuscript, Line 110-112, page 3)

  1. Lines 107-108: Please, indicate the well size for this 48-well plate.

Answer: Thank you for your valuable suggestion. We have described the dimensions of the well in the 48-well plate. (Revised manuscript, Line 113-114, page 3)

  1. Line 110: Was it the same type of 24-well insect box that was used for the mortality tests (lines 170-171)? If yes, the well size should be indicated here (not below, in line 171). If not, please, indicate the well size for these boxes.

Answer: Thank you for your valuable suggestion. The 24-well plate sizes used in this study are all the same, and we have added a note on the 24-well plate sizes here. (Revised manuscript, Line 116-117, page 3)

  1. Lines 146, 159 and 169: for feeding preference tests, “same-sized second instar larvae” were used, whereas to estimate the mortality, “late second instar larvae with the same body weight” were used. Please, explain the reasons of this difference (size vs. weight) and also indicate approximated mean size and weight (or their variation ranges).

Answer: Thank you for your valuable suggestion. We are very sorry thatthis is a mistake in our writing. In fact, in the feeding preference and mortality, as well as in the subsequent growth, development and reproduction experiments, the second instar larvae with a body weight of 6-8mg were selected. (Revised manuscript, Line 149-152, Line 162-165, Line 176, page 4)

  1. Line 152: Possibly, the leaf discs were not photographed but scanned (as in line165)? If not, why in the same line “the scanning resolution” is mentioned? If they were really photographed, why two different methods for square estimation were used for very similar tests?

Answer: Thank you for your valuable suggestion. We are very sorry for the confusion caused to you by the wrong description of the experimental method. In fact, we first use the camera to take pictures of the leaves, and then calculate the feeding area by Adobe Photoshop 2020 software. (Revised manuscript, Line 156-158, Line 171-172, page 4)

  1. Line 171: Delete superscript “3” after “cm” – size is measured in linear, not in cubic centimeters.

Answer: Thank you for your valuable suggestion. We have implemented corresponding changes to this part of the content in the revised manuscript. (Revised manuscript, Line 114 and 117, page 3)

  1. Lines 174-175: “Each experiment was conducted with 5 replicates per treatment, resulting in 20 replicates for each treatment” – this sentence is confusing. Possibly, some words were lost, such as “was conducted 4 times with 5 replicates”?

Answer: Thank you for your valuable suggestion. We apologize for the error in the description of the relevant content. The experimental treatment should be "Each cotton variety was subjected to five replicate experiments, with each replicate consisting of 20 larvae". (Revised manuscript, Line 180-181, page 4)

  1. Lines 180-182: Was the weight of larvae recorded every 24 h (line 181) or only on the day when the larvae entered a new instar (line 182)?

Answer: Thank you for your valuable suggestion. We apologize for the error in the description of the relevant content. We observed and recorded the instars of H.armigera larvae in each group every 24 h. On the day when H.armigera entered the new instar, the weight of H.armigera larvae was recorded. (Revised manuscript, Line 186-188, page 4)

  1. Line 182-183: Incomplete sentence (the verb is absent). Possibly, it is the beginning of the next one?

Answer: Thank you for your valuable suggestion. We apologize for not being clear about the relevant content. We have made changes to the relevant content. (Revised manuscript, Line 188-193, page 4)

  1. Line 199: WHAT «controlled conditions”? Please, indicate clearly: temperature, humidity, etc.

Answer: Thank you for your valuable suggestion. We have added the control conditions to the revised manuscript. (Revised manuscript, Line 204-205, page 5)

  1. Lines 218-219, equation (6): are you sure that namely “pupation rate” in percents (not the total number of pupae) was used to calculate the eclosion rate?

Answer: Thank you for your valuable suggestion. Sorry, this is a translation error and should be calculated using the total number of pupae in each repeat. We have made changes to the relevant content. (Revised manuscript, Lines 228, equation (6), page 5)

  1. Lines 218-219, equations (3)-(6): What you mean “total number”: total per replicate or total per the whole experiment? This should be clearly indicated.

Answer: Thank you for your valuable suggestion. The formula refers to the total number of individuals in each repetition. We have modified these equations more explicitly. (Revised manuscript, Lines 228, equation (2)-(9), page 5-6)

  1. General comment to figures: the graphs are too small and of rather low quality. In particular, it is practically impossible to read the words and numbers along the vertical axis and the letters above the bars.

Answer: Thank you for your valuable suggestion. We are very sorry, we submitted manuscripts are high-definition images, but due to some unknown reasons, resulting in reduced image quality. The image has been re-uploaded to a higher definition version in the form of an attachment, and the data in the chart (mean, standard deviation, and significance) has been supplemented in the form of an attached table. (Supplementary File, page 1-5; Figures, Figure 1-4)

  1. Figure 1-III–What measure of variation is shown for the percentage of mortality?

Answer: Thank you for your valuable suggestion. We use a brush to touch the cotton bollworm gently. If it does not move, it is judged that it has died. The ordinate is the mortality rate ( % ), and the abscissa is the treatment group of different varieties of cotton, where A, B, C, D, E represent CCRI 49, Chuangmian 508,Tahe 2, J206-5, Zhongzhimian KV2, respectively. Mean, standard deviation and significance are shown in the figure. Different lowercase letters indicated that there were significant differences among different varieties by Duncan multiple comparison (P < 0.05). In the figure note section, we have made a more clear exposition. Due to previous writing errors, all charts in this article use standard deviation. If we do not understand your suggestion, please feel free to contact us. (Revised manuscript, Lines 255-264, page 6-7, Supplementary File, page 1; Figures, Figure 1)

  1. Line 257: This sentence is a bit confusing: “no significant difference between the third and fourth instar larvae” means that these two instars were of the same mean weight? This is just impossible (see lines 261-262). Possibly, it is a misprint and the comma after “larvae” should be replaced by “from” or “among”?

Answer: Thank you for your valuable suggestion. This part is due to the problem of we translation, so you are confused about this content, for which We are sorry. We have made changes to the relevant content. (Revised manuscript, Lines 267-273, page 7)

  1. Figure 2: What measure of variation is shown for weight and duration? Is it SE as in the first figure”. In any case, this information should be included in the legend.

Answer: Thank you for your valuable suggestion. We are very sorry, we submitted manuscripts are high-definition images, but due to some unknown reasons, resulting in reduced image quality. The image has been re-uploaded to a higher definition version in the form of an attachment, and the data in the chart (mean, standard deviation, and significance) has been supplemented in the form of an attached table. Mean, standard deviation and significance are shown in the figure. Different lowercase letters indicated that there were significant differences among different varieties by Duncan multiple comparison (P < 0.05). Due to previous writing errors, all charts in this article use standard deviation. If we do not understand your suggestion, please feel free to contact us. (Revised manuscript, Lines 305-321, page 8, Supplementary File, page 1-3; Figures, Figure 2)

  1. Figures 3 and 4: Again, what measure of variation is shown? Is it the same for duration (days), weight (mg), rates (%), etc.?

Answer: Thank you for your valuable suggestion. We are very sorry, we submitted manuscripts are high-definition images, but due to some unknown reasons, resulting in reduced image quality. The image has been re-uploaded to a higher definition version in the form of an attachment, and the data in the chart (mean, standard deviation, and significance) has been supplemented in the form of an attached table. Mean, standard deviation and significance are shown in the figure. Different lowercase letters indicated that there were significant differences among different varieties by Duncan multiple comparison (P < 0.05). Due to previous writing errors, all charts in this article use standard deviation. If we do not understand your suggestion, please feel free to contact us. (Revised manuscript, Lines 355-369, page 9, Supplementary File, page 3-4; Figures, Figure 3)

  1. Line 418: Spodoptera exigua should be in the Italics font.

Answer: Thank you for your valuable suggestion. We have made changes to the relevant content. (Revised manuscript, Lines 455, page 11)

  1. Line 421: I suggest that the text starting from “In summary..” should be separated in a new paragraph.

Answer: Thank you for your valuable suggestion. We have modified the logical structure of this part of the content to make it clearer. (Revised manuscript, Lines 453-464, page 11)

In addition, at the end of the discussion section of this paper, we elaborate on the limitations of this study and put forward the prospect of future research directions. (Revised manuscript, Lines 465-475, page 11-12)

  1. Lines 424, 425, etc.: S. frugiperda, S. inferens, etc. should be in the Italics font. Please, check carefully all Latin names in the manuscript.

Answer: Thank you for your valuable suggestion. We have made changes to the relevant content. (Revised manuscript, Lines 457-459, page 11)

  1. Lines 433-462: This is too long text for the Conclusions that are expected to be several short but informative sentences. Please, consider to make it shorter.

Answer: Thank you for your valuable suggestion. We make a more brief summary of the content of the conclusion. (Revised manuscript, Lines 477-485, page 12)

  1. Line 473: Separate the words “The corresponding”

Answer: Thank you for your valuable suggestion. We have made changes to the relevant content. (Revised manuscript, Lines 496-497, page 12)

Reviewer 2 Report

Comments and Suggestions for Authors

Yue et al. report the results of a laboratory study on the feeding response of Helicoverpa armigera larvae to five varieties of cotton from China.

I was unable to review this manuscript because no information was provided on the statistical treatment of results and the figures were unreadable.

MAJOR POINTS.

(I) It was impossible to know whether the data had been analyzed correctly as no details of the statistical procedures were provided.

(II) All the the figures were unreadable. It was therefore impossible to know whether the text description of the results agreed with the results shown in the figures.

(III) The Discussion focused on issues that were not the subject of the study. I found no focus on feeding behavior, or the factors that affect lepidopteran feeding responses, the chemical or physical differences among the varieties tested, or information on susceptibility to other pests.  The authors need to read the extensive literature on this subject.

I wrote numerous suggestions and numbered points on a scanned copy of the manuscript.

NUMBERED POINTS (see scanned file)

1. Only one author is marked #, not two.

2. Forgive my ignorance, please indicate whether Xinjiang is a province, region, municipality, state, city or something else.

3. Surely the conclusion of the Abstract should be that field testing is required to validate the laboratory findings on feeding preferences.

4. Do not use keywords that are already in the title.

5. The international unit for agriculture area is "hectares" (ha). Please convert mu to ha throughout the manuscript.

6. Please given percentages without decimal places.

7. What is lodging resistance? Please reword or explain.

8. Give full name at first use.

9. The usual abbreviation for dollars is "USD $"

10. Suggest you change this to 1.5 – 3.3 billion yuan.

11. Given species name of whitefly.

12. excavated? Do you mean evaluated?

13. Please provide a list of the ingredients and quantities used for the insect diet. This is important because insect diet affects subsequent feeding preferences.

14. Provide details of ALL treatments applied to cotton prior to their use in experiments (this can be supplementary material).

15. Text is unnecessary and should be deleted.

16a. Do you mean that larvae were starved for 4 h prior to the start of the experiment? Or please explain.

16b. How often were larvae checked to see whether the leaf disc had been completely eaten prior to the end of the 24 h period?

17. Supplemented? Please reword or explain.

18a. What was the mean ±SE body weight of second instars?

18b. How many larvae were used in each box? 24 larvae?

19. A well size cannot be 2 x 2 x 2.5 cm3 – this does not make sense. Do you mean 2 x 2 x 2.5 cm?

20. I did not understand. Were there 5 or 20 replicates of each treatment?

21. This is not a sentence. Please reword.

22. Delete obvious text.

23. Do you mean the percentage or proportion of deformed pupae?

24. Do you mean pupal sex ratio?

25. Do you mean the percentage or proportion of adult emergence?

26. Do you mean the percentage or proportion of deformed adults of each sex?

27. Delete this repetitive text.

28. Please provide information on your statistical analyses.

If parametric analyses were used how did you check for assumptions such as equality of variances?

29. How were treatments compared statistically?

30. What do these probabilities refer to? No details of statistical tests have been given.

The authors should fully disclose the stats, such as F values, degrees of freedom and P values.

31. Means should be given with their corresponding SE or SD values.

32. I found it impossible to read this figure as the text is tiny and the resolution is low. MDPI states that figures should be 600 ppi resolution.

Increase the size of figure 1 and use larger text.

33. Again, the figure 2 is not readable and has almost microscopic text.

34. Figure 3 is unreadable.

35. Figure 4 is unreadable.

36. Discussion. You have already mentioned the economic importance of H. armigera in the Introduction.

37. This long text on transgenics is not relevant as you did not test transgenic varieties.

38. This is a reiteration of results, not discussion.

39. A conclusion should be a brief (three sentences) take-home message from the study.

40. The Acknowledgements appear to be written by a student, not by the SEVEN authors of the study.

References: there are formatting errors and typos in the references. I only checked the first page.

Comments on the Quality of English Language

See my suggestions for improvements.

Author Response

We sincerely thank you for your valuable comments and suggestions. In response to the problems you pointed out, we have made corresponding revisions. If there is any deviation in the process of understanding your amendment, or if the amendment content does not accurately reflect your intention, please feel free to contact us.

  1. 1.Only one author is marked #, not two.

Answer: Thank you for your valuable suggestion. We have made changes to the relevant content. (Revised manuscript, Lines 4, page 1)

  1. Forgive my ignorance, please indicate whether Xinjiang is a province, region, municipality, state, city or something else.

Answer: Thank you for your valuable suggestion. Xinjiang is a provincial-level administrative region in China. Its full name is Xinjiang Uyghur Autonomous Region. It is the main cotton producing area in China.

  1. Surely the conclusion of the Abstract should be that field testing is required to validate the laboratory findings on feeding preferences.

Answer: Thank you for your valuable suggestion. We have made changes to the relevant content. (Revised manuscript, Lines 42-44, page 1)

  1. Do not use keywords that are already in the title.

Answer: Thank you for your valuable suggestion. We changed the keywords. (Revised manuscript, Lines 45, page 1)

  1. The international unit for agriculture area is "hectares" (ha). Please convert mu to ha throughout the manuscript.

Answer: At the suggestion of another reviewer, we have removed this section that deals with the yield of various cotton varieties and their producers. We would like to express our sincere thanks to you for your revision. We will pay more attention to the use of international units in my future writing. (Revised manuscript, Lines 48-55, page 2)

  1. Please given percentages without decimal places.

Answer: Thank you for your valuable suggestion. We have omitted the number after the decimal point. (Revised manuscript, Lines 52-54, page 2)

  1. What is lodging resistance? Please reword or explain.

Answer: Thank you for your suggestion. The term "lodging resistance" refers to the ability of crops to maintain an upright growth posture when faced with environmental challenges such as wind, rain, and waterlogging.

  1. Give full name at first use.

Answer: Thank you for your valuable suggestion. We have made changes to the relevant content. (Revised manuscript, Lines 56, page 2)

  1. The usual abbreviation for dollars is "USD $"

Answer: Thank you for your valuable suggestion. We have made changes to the relevant content. (Revised manuscript, Lines 62, page 2)

  1. Suggest you change this to 1.5 – 3.3 billion yuan.

Answer: Thank you for your valuable suggestion. This section of currency has been converted to US dollars. (Revised manuscript, Lines 65, page 2)

  1. Given species name of whitefly.

Answer: Thank you for your valuable suggestion. We added the Latin name of whitefly to the corresponding section. (Revised manuscript, Lines 77-78, page 2)

  1. excavated? Do you mean evaluated?

Answer: Thank you for your valuable suggestion. The objective of this section is to identify an exemplary variety with robust resistance to insects. We have made changes to the relevant content. (Revised manuscript, Lines 91-96, page 2)

  1. Please provide a list of the ingredients and quantities used for the insect diet. This is important because insect diet affects subsequent feeding preferences.

Answer:Thank you for your valuable suggestion. We have added the composition and ratio of insect feed to the corresponding position in the article. (Revised manuscript, Lines 101-103, page 3)

  1. Provide details of ALL treatments applied to cotton prior to their use in experiments (this can be supplementary material).

Answer: Thank you for your valuable suggestion. We have added this part to the supplementary material. The five cotton varieties in the experiment were sown in April 2024, and planted in 20 × 20 × 15 cm flowerpots, each with 1 plant. The seed sowing depth was 3 cm, and the whole growth cycle of cotton was controlled at 15-35°C. Watering once every 5 days, 3 L water for each flower pot. Before the emergence of true leaves of cotton, according to the use of mepiquat chloride : water = 0.2 g : 1 kg, the growth and development of the above-ground part was controlled. After the cotton bud appeared, according to the proportion of nitrogen, phosphorus and potassium = 1: 0.3: 1 fertilization once. No insecticides, herbicides and fungicides were applied during the test period.

  1. Text is unnecessary and should be deleted.

Answer: Thank you for your valuable suggestion. We have deleted this part of the content. (Revised manuscript, Lines 134, page 3)

16a. Do you mean that larvae were starved for 4 h prior to the start of the experiment? Or please explain.

Answer: Yes.

16b. How often were larvae checked to see whether the leaf disc had been completely eaten prior to the end of the 24 h period?

Answer: Thank you for your valuable suggestion. We check every 8 hours.

  1. Supplemented? Please reword or explain.

Answer: Thank you for your valuable suggestion. This part means that when the leaves are completely eaten by the larvae within 24 hours, new leaves of the same variety are immediately added.

18a. What was the mean ±SE body weight of second instars?

Answer: Thank you for your valuable suggestion. We are very sorry that due to the inaccuracy of the previous description, the body weight of the second instar larvae selected in the experiment was between 6-8mg, and the mean ±SE was not calculated. We have made changes to the relevant content. (Revised manuscript, Lines 149-152, page 4)

18b. How many larvae were used in each box? 24 larvae?

Answer: Thank you for your valuable suggestion. We placed 20 larvae in each 24-well plate. We have made changes to the relevant content. (Revised manuscript, Lines 180-181, page 4)

  1. A well size cannot be 2 x 2 x 2.5 cm3 – this does not make sense. Do you mean 2 x 2 x 2.5 cm?

Answer: Yes, we have made changes to the relevant content. (Revised manuscript, Lines 117, page 3)

  1. I did not understand. Were there 5 or 20 replicates of each treatment?

Answer: Thank you for your valuable suggestion. We apologize for the error in the description of the relevant content. The experimental treatment should be "Each cotton variety was subjected to five replicate experiments, with each replicate consisting of 20 larvae". (Revised manuscript, Line 180-181, page 4)

  1. This is not a sentence. Please reword.

Answer: Thank you for your valuable suggestion. We have modified this part. (Revised manuscript, Line 186-193, page 4)

  1. Delete obvious text.

Answer: Thank you for your valuable suggestion. The previous expression is wrong, we have modified this part. (Revised manuscript, Line 186-193, page 4)

  1. Do you mean the percentage or proportion of deformed pupae?

Answer: Thank you for your valuable suggestion. The previous expression is wrong, we have modified this part. (Revised manuscript, Line 186-193, page 4)

  1. Do you mean pupal sex ratio?

Answer: Yes, We have modified this part. (Revised manuscript, Line 186-193, page 4)

  1. Do you mean the percentage or proportion of adult emergence?

Answer: Thank you for your valuable suggestion. The previous expression is wrong, we have modified this part. (Revised manuscript, Line 186-193, page 4)

  1. Do you mean the percentage or proportion of deformed adults of each sex?

Answer: Thank you for your suggestion. The previous description of this part was inaccurate and has been modified. (Revised manuscript, Line 186-193, page 4)

  1. Delete this repetitive text.

Answer: Thank you for your suggestion. We have modified this part of the content. (Revised manuscript, Line 186-193, page 4)

  1. Please provide information on your statistical analyses.

If parametric analyses were used how did you check for assumptions such as equality of variances?

Answer: Thank you for your suggestion. We have modified this part of the content. After the preliminary analysis of the data through the calculation formula in the article, the SPSS 20.0 software was used to perform one-way analysis of variance, and the Duncan test was used for multiple comparisons to determine whether there were significant differences in multiple sets of samples. Then use Origin 2017 software to draw the analysis results. Different lowercase letters indicated that there were significant differences among different varieties by Duncan multiple comparison (P < 0.05). (Revised manuscript, Line 212-218, page 5)

  1. How were treatments compared statistically?

Answer: Thank you for your suggestion. We have modified this part of the content. The means of body weight, development period, mortality, eclosion rate and other data were calculated by formula. Then one-way analysis of variance was performed by spss software, and Duncan test was used for multiple comparisons to analyze whether the differences between different treatment groups were significant. We have modified these equations more explicitly. (Revised manuscript, Lines 228, equation (2)-(9), page 5-6)

  1. What do these probabilities refer to? No details of statistical tests have been given.

The authors should fully disclose the stats, such as F values, degrees of freedom and P values.

Answer: “P < 0.05” meant that there was significant difference among different treatment groups. “P > 0.05” indicated no significant difference between different treatment groups. The data in the figure (mean , standard deviation and significance) have been supplemented in tabular form.

  1. Means should be given with their corresponding SE or SD values.

Answer: The data in this paper are in the form of standard deviation. The standard deviation data of the mean in the paper has been supplemented by the tabular form.

  1. I found it impossible to read this figure as the text is tiny and the resolution is low. MDPI states that figures should be 600 ppi resolution.Increase the size of figure 1 and use larger text.

Answer: Thank you for your valuable suggestion. We are very sorry, we submitted manuscripts are high-definition images, but due to some unknown reasons, resulting in reduced image quality. The image has been re-uploaded to a higher definition version in the form of an attachment, and the data in the chart (mean, standard deviation, and significance) has been supplemented in the form of an attached table. (Supplementary File, page 1-5; Figures, Figure 1-4).

  1. Again, the figure 2 is not readable and has almost microscopic text.

Answer: Thank you for your valuable suggestion. The image has been re-uploaded to a higher definition version in the form of an attachment, and the data in the chart (mean, standard deviation, and significance) has been supplemented in the form of an attached table. (Revised manuscript, Lines 305-321, page 8, Supplementary File, page 1-3; Figures, Figure 2)

  1. Figure 3 is unreadable.

Answer: Thank you for your valuable suggestion. The image has been re-uploaded to a higher definition version in the form of an attachment, and the data in the chart (mean, standard deviation, and significance) has been supplemented in the form of an attached table. (Revised manuscript, Lines 355-369, page 9, Supplementary File, page 3-4; Figures, Figure 3)

  1. Figure 4 is unreadable.

Answer: Thank you for your valuable suggestion. The image has been re-uploaded to a higher definition version in the form of an attachment, and the data in the chart (mean, standard deviation, and significance) has been supplemented in the form of an attached table. (Revised manuscript, Lines 393-404, page 10, Supplementary File, page 4-5; Figures, Figure 4)

  1. Discussion. You have already mentioned the economic importance of H. armigerain the Introduction.

Answer: Thank you for your valuable suggestion. We have deleted this part of the content

  1. This long text on transgenics is not relevant as you did not test transgenic varieties.

Answer: Thank you for your valuable suggestion. We have deleted this part of the content

  1. This is a reiteration of results, not discussion.

Answer: Thank you for your valuable suggestion. We have modified this part of the content. (Revised manuscript, Lines 406-475, page 12)

  1. A conclusion should be a brief (three sentences) take-home message from the study.

Answer: Thank you for your valuable suggestion. We make a more brief summary of the content of the conclusion. (Revised manuscript, Lines 477-485, page 12)

  1. The Acknowledgements appear to be written by a student, not by the SEVEN authors of the study.

Answer: Thank you for your valuable suggestion. We have modified this part of the content. (Revised manuscript, Lines 498-499, page 12)

  1. References: there are formatting errors and typos in the references. I only checked the first page.

Answer: Thank you for your valuable suggestion. We have modified this part of the content. (Revised manuscript, Lines 502-561, page 12-13)

Round 2

Reviewer 2 Report

Comments and Suggestions for Authors

The authors have made various improvements but they have not addressed as series of my previous requests:

1. L46-48: please indicate whether Xinjiang is a province, region, municipality, state, city or something else.

2. Surely the conclusion of the Abstract should be that field testing is required to validate the laboratory findings on feeding preferences.

3. L162 – should read "...20 larvae in each replicate...."

L148, L161 – move the phrase on starved larvae to the beginning of the sentences "Larvae that had been starved for 4 h were placed in the middle of each Petri dish....."

4. If parametric analyses were used how did you check for assumptions such as equality of variances?

The same question applies to checks on the normality of data prior to ANOVA.

5. The authors should fully disclose the stats, i.e., F values, degrees of freedom and P values. Showing P values is not enough.

If the authors do not wish to disclose their F statistics in the text, they should present the FULL anova tables in the supplementary material.

6. Again, all the figures have microscopic text that cannot be read. Please increase the size of text on axes and axes labels and letters above columns in graphs to make the figures readable.

7. The authors have not addressed my previous request to present percentages without decimal places in sections 3.3, 3.4 and the Discussion. You only used 100 larvae or less, so presenting percentages with one or two decimal places implies that you used thousands of insects, which you did not.

8. L371 – what does eggs/head mean? Please reword for clarity.

9. Errors persist in the formatting of the references section.

Author Response

Response to Reviewer

We would like to thank you for your professionalreview work,constructive comments, and valuablesuggestions on our manuscript. Your time and effortsare greatly appreciated. We have strived to improvethe manuscript accordingly as listed in details below. Should there be any shortcomings within the content, you are kindly requested to notify us promptly, thereby enabling us to undertake further enhancements.

  1. L46-48: please indicate whether Xinjiang is a province, region, municipality, state, city or something else.

Answer: Thank you for your valuable suggestion. According to your suggestion, we have revised the corresponding part of the article. (Revised manuscript, Line 48-50, page 2)

  1. Surely the conclusion of the Abstract should be that field testing is required to validate the laboratory findings on feeding preferences.

Answer: Thank you for your valuable suggestion. According to your suggestion, we have revised the corresponding part of the article. (Revised manuscript, Line 42-45, page 1)

  1. L162-should read "...20 larvae in each replicate...."

L148, L161-move the phrase on starved larvae to the beginning of the sentences "Larvae that had been starved for 4 h were placed in the middle of each Petri dish....."

Answer: Thank you for your valuable suggestion. According to your suggestion, we have revised the corresponding part of the article. (Revised manuscript, Line 149-152, Line 163-166, page 4)

  1. If parametric analyses were used how did you check for assumptions such as equality of variances?

The same question applies to checks on the normality of data prior to ANOVA.

Answer: Thank you for your valuable suggestion. We employed the Shapiro-Wilk test to ascertain the normality of the data distribution. The P-value of the Shapiro-Wilk Test is greater than 0.05, indicating that the data conforms to a normal distribution. Subsequently, we employ the Levene test to evaluate the homogeneity of variances. The P-value of Levene test is greater than 0.05, which indicates that homogeneity test of variances is satisfied, meaning the variances are equal. Consequently, one-way analysis of variance can be conducted.

  1. The authors should fully disclose the stats, i.e., F values, degrees of freedom and P values. Showing P values is not enough.

If the authors do not wish to disclose their F statistics in the text, they should present the FULL anova tables in the supplementary material.

Answer: Thank you for your valuable suggestion. We have provided data such as F values, degrees of freedom, etc. in the supplementary material. (Supplementary Material, page 1-6)

  1. Again, all the figures have microscopic text that cannot be read. Please increase the size of text on axes and axes labels and letters above columns in graphs to make the figures readable.

Answer: Thank you for your valuable suggestion. According to your requirements, we have redrawn the drawings in the manuscript and uploaded the original drawings in high definition through the compressed package. (Revised manuscript, Lines 253, Figure 1, page 6; Lines 303, Figure 2, page 8; Lines 353, Figure 3, page 9; Lines 392 Figure 4, page 10).

Furthermore, we have incorporated several original high-resolution images into our response. (Response to Reviewer, page 3-6)

  1. The authors have not addressed my previous request to present percentages without decimal places in sections 3.3, 3.4 and the Discussion. You only used 100 larvae or less, so presenting percentages with one or two decimal places implies that you used thousands of insects, which you did not.

Answer: Dear reviewer, thank you for your valuable suggestions. The experiments conducted in sections 3.3 and 3.4 of this study were sequentially executed subsequent to the mortality experiment. Consequently, the denominator utilized for calculating indices such as the pupal malformation rate, female ratio, and eclosion rate (refer to equations (3)-(6), page 5-6) is not the aggregate count of 100 larvae. In order to portray the calculations with greater precision, we suggest retention of two decimal places.

  1. L371-what does eggs/head mean? Please reword for clarity.

Answer: Thank you for your valuable suggestion. The figure here refers to the average number of eggs laid per female. According to your suggestion, we have revised the corresponding part of the article. (Revised manuscript, Line 377-379, page 10)

  1. Errors persist in the formatting of the references section.

Answer: Thank you for your valuable suggestion. According to your suggestion, we have revised the corresponding part of the article. (Revised manuscript, Line 503-601, page 12-14)

Round 3

Reviewer 2 Report

Comments and Suggestions for Authors

The authors have addressed most of my concerns.

Author Response

We would like to express our deepest gratitude to you for your valuable advice on our manuscript. Your review not only accurately pointed out flaws that we had not noticed, but also patiently suggested practical improvements. Your careful comments have played a vital role in the improvement of our document.